# Persistence as a Constituent of a Biocontrol Mechanism (Competition for Nutrients and Niches) in *Pseudomonas putida* PCL1760

**DOI:** 10.3390/microorganisms11010019

**Published:** 2022-12-21

**Authors:** Aynur Kamilevich Miftakhov, Roderic Gilles Claret Diabankana, Mikhail Frolov, Marat Miratovich Yusupov, Shamil Zavdatovich Validov, Daniel Mawuena Afordoanyi

**Affiliations:** 1Laboratory of Molecular Genetics and Microbiology Methods, Kazan Scientific Center of Russian Academy of Sciences, 420111 Kazan, Russia; 2Centre of Agroecological Research, Kazan State Agrarian University, 420015 Kazan, Russia; 3FRC Kazan Scientific Center, Tatar Scientific Research Institute of Agricultural Chemistry and Soil Science, Russian Academy of Sciences, 420111 Kazan, Russia

**Keywords:** *Pseudomonas putida*, biocontrol mechanism, persistence, *rsf*S gene, competition for nutrients and niches, root colonizers, *Fusarium oxysporum*, *rsf*S gene knockout mutant

## Abstract

Competition for nutrients and niches (CNN) is known to be one of the mechanisms for biocontrol mostly exhibited by *Pseudomonas* strains. Phenotypic and full genome analysis revealed *Pseudomonas putida* PCL1760 controlling tomato foot and root rot (TFRR) solely through CNN mechanism. Although the availability of nutrients and motility are the known conditions for CNN, persistence of bacteria through dormancy by ribosomal hibernation is a key phenomenon to evade both biotic and abiotic stress. To confirm this hypothesis, *rsf*S gene knockout mutant of PCL1760 (SB9) was first obtained through genetic constructions and compared with the wild type PCL1760. Primarily, relative expression of *rsf*S in PCL1760 was conducted on tomato seedlings which showed a higher expression at the apical part (1.02 ± 0.18) of the plant roots than the basal (0.41 ± 0.13). The growth curve and persistence in ceftriaxone after the induction of starvation with rifampicin were performed on both strains. Colonization on the tomato root by CFU and qPCR, including biocontrol ability against *Fusarium,* was also tested. The growth dynamics of both PCL1760 and SB9 in basal and rich medium statistically did not differ (*p* ≤ 0.05). There was a significant difference observed in persistence showing PCL1760 to be more persistent than its mutant SB9, while SB9 (pJeM2:*rsf*S) was 221.07 folds more than PCL1760. In colonization and biocontrol ability tests, PCL1760 was dominant over SB9 colonizing and controlling TFRR (in total, 3.044 × 10^4^ to 6.95 × 10^3^ fg/µL and 55.28% to 30.24%, respectively). The deletion of the *rsf*S gene in PCL1760 reduced the persistence and effectiveness of the strain, suggesting persistence as one important characteristic of the CNN.

## 1. Introduction

As an environmentally friendly practice and to attain safe agricultural products for consumers, the biological approach to plant protection is attracting considerable interest worldwide. The world market for biopreparations used in protecting agricultural plants is increasing yearly, reaching USD 1 billion by 2025 [1]. Since the last decades, growing application of microbes for crop protection by new biopreparations is being developed through profound knowledge of the mechanisms for plant–microbial interaction. As well as *Bacillus* strains, *Pseudomonas* species produce volatile bioactive compounds involved in parasitism and predation, antibiosis, competition for nutrients and niches (CNN) and induce systemic resistance (ISR) for biocontrol [2,3]. Among these mechanisms, CNN was initially considered as a delivery tool for strains acting via antibiosis, induced systemic resistance (ISR), or parasitism and predation [4].

Hitherto, conditions necessary for the mechanism of CNN include the availability of root exudates by the plant and the structures for the mobility of the bacteria. Two strains previously isolated using enrichment on tomato root tip, *Pseudomonas fluorescens* PCL1751 and *P. putida* PCL1760, were proven to protect tomato plants against *Fusarium* root rot via the CNN [5,6]. With the exception of protease secretion in PCL1751, both strains do not produce toxic metabolites, lytic enzymes, or trigger ISR [5,6]. The non-motile mutant of the *P. fluorescens* strain PCL1751 and mutant of the *P. putida* strain PCL1760 impaired in dicarboxylic acids consumption demonstrated a less efficient control of *Fusarium* root rot on tomato, thus confirming the above conditions [5,7]. Correspondingly, a full genome analysis of the *P. putida* strain PCL1760 revealed the absence of genetic systems for toxic secondary metabolites, while PCL1751, which phenotypically showed no antifungal activity, had these cluster genes present [5,8,9,10]. This confirms CNN as the sole mechanism of these two strains to suppress TFRR. We suggest the persistence of a bacteria strain to be the third condition, in addition to the two above-mentioned conditions, as an important property for a good root colonizer. The viability and persistence of strains to control pathogens in adverse conditions depend on energy conservation and acquisition controlled by the “fine-tuning” of regulatory factors involved in cell hibernation (starvation) and activation (nutrient availability), respectively.

Hibernation in bacteria occurs during the stationary phase with a programmed disassembly of ribosomes to conserve energy expense on protein biosynthesis [11]. Another important advantage of hibernation in the stationary phase is to resist xenobiotics that inhibit bacterial growth which can be considered as the persistence of bacteria to antibiotics [12]. This is an important trait in biocontrol agents to resist xenobiotics produced by other bacteria as a form of antibiosis against susceptible strains. A review article on the hibernation stage of ribosomes in bacteria by Usachev and colleagues [13] thoroughly explains the importance, mechanism, and regulatory factors involved in ribosome assembly and disassembly. The authors made mention of the key factors and the role they play in hibernation, including the ribosome modulation factor (RMF), hibernation protein factors (HPFs), the ribosome-associated inhibitor YfiA (RaiA), and the ribosome silencing factor RsfS (RsfA and YbeB) which is mostly present in eubacteria [14]. The ribosomal silencing factor S (RsfS) is a stationary phase protein that binds to uL14 on the ribosome large subunit (50S), preventing subunit association and formation of the 70S initiating complex, thus suppressing protein synthesis under stressful conditions [15]. Interestingly, the RsfS is not just a hibernation factor but also a biogenesis factor with similarity to the mitochondrial ortholog MALSU1 in eukaryotic cells [16].

Hereinafter, considering the above-mentioned mechanism, the CNN, we propose a key phenotypic characteristic of a biological control agent acting through CNN to include also the persistence of xenobiotics. In the present work, we obtained a mutant (SB9) of the *P. putida* strain PCL1760 with a ribosome silencing factor S (*rsf*S) knocked out. We then compared the phenotypic characteristics of PCL1760 to its mutant (SB9) to confirm their impaired properties.

## 2. Materials and Methods

### 2.1. Strains and Growth Conditions

Microbial strains used in this study are listed in Table 1. *Escherichia coli* were cultivated on LB medium at 37 ± 1 °C. *Pseudomonas putida* PCL1760 and its derivatives were grown at 30 ± 1 °C in a (SSM) standard succinate medium [17] to prevent a spontaneous mutation to auxotrophy. Both media were amended with kanamycin to a final concentration of 50 µg/mL when needed.

### 2.2. The Mating Procedure and Generation of a PCL1760 (ΔrsfS) Mutant

The gene-specific oligonucleotide primers used in this study are listed in Table 2. All primers were designed based on the full genome of *P. putida* PCL1760 using Clone Manager 9 (Sci Ed Central, Cary, NC, USA).

The *rsf*S gene deletion mutant *P. putida* PCL1760 was obtained by homologous recombination [22], as presented in Figure 1 and Figure 2:

#### 2.2.1. Amplification of *rsf*S Gene Fragments with Flanking Sequences

The fragment of the *rsf*S gene with flanking sequences (fl-rsfS) was amplified by PCR from the chromosomal DNA of *P. putida* PCL1760. For this purpose, a primer fl-rsfS located on both sides (5′ and 3′) of the target gene at a distance of ~1000 bp (Figure 1) was designed based on the full genome of *P. putida* PCL1760 [9] using Clone Manager 9 (Sci Ed Central, Cary, NC, USA).

#### 2.2.2. The Insertion of Target Gene Fragments with Flanking Sequences into the pUC19 Vector

The restriction digest of the pUC19 vector was carried out at the SmaI restriction site. Ligation of the pUC19 vector with fl-rsfS (pUC19:fl-rsfS) was performed using T4 DNA ligase (Evrogen, Moscow, Russia), then pretreated with a SmaI restriction enzyme once again to get rid of the vector molecules without insertion. Restriction products were further separated in 1% agarose gel electrophoresis and extracted from the gel using a Cleanup Mini standard kit (Evrogen, Moscow, Russia) according to the manufacturer’s instructions. The construct was PCR-amplified and purified using a Plasmid Miniprep standard kit (Evrogen, Moscow, Russia).

#### 2.2.3. Deletion of the *rsf*S Gene in the pUC19: fl-rsfS Vector

The deletion of the *rsf*S gene in the pUC19: fl-rsfS vector was performed by inverse PCR, using the inverse *rsf*S primer (inv- *rsf*S) of the target fragment [23]. The PCR product (pUC19: flΔ*rsf*S) was treated with the restriction enzyme MalI, separated in 1% agarose gel electrophoresis, and then extracted from the gel using a Cleanup Mini standard kit (Evrogen, Moscow, Russia), according to the manufacturer’s instructions.

#### 2.2.4. The Insertion of Target Gene Fragments with flΔrsfS Flanking Sequences into the pK18mobsacB Vector

The restriction digest of pUC19: flΔ*rsf*S and pK18mobsacB vectors was carried out at the *Eco*RI and *Bam*HI restriction sites. After the restriction digest of pUC19:flΔ*rsf*S and pK18mobsacB vectors, and ligation of flΔ*rsf*S and pK18mobsacB fragments, a new construct pK18mobsacB:flΔ*rsf*S was obtained.

#### 2.2.5. Transformation of Plasmid Vectors

A volume of 100 µL of competent cells was added into a cold sterile 1.5 mL microcentrifuge tube and thawed on ice. A total of 10 ng (10 µL) of a DNA vector was added to the suspension, gently mixed, and kept on ice for 20 min with a subsequent heat shock at 42 °C for 1 min. Next, 900 μL of sterile LB was added into a tube at room temperature. The suspension was incubated for 1 h at 37 ± 1 °C, after which 100 μL was plated on Lysogeny broth medium (LB) [g/L: tryptone,10 g; yeast extract, 5 g; NaCl, 10 g] amended with kanamycin to a final concentration of 50 µg/mL, as a selective medium, and incubation was at 37 °C.

#### 2.2.6. Conjugative Transfer of Plasmid Vectors from *E. coli* S17-1 to *P. putida* Strains

*E. coli* strain S17-1 was transformed using the pK18mobsacB: flΔ*rsf*S vector. The cells of *P. putida* PCL1760 and *E. coli* S17-1×pK18mobsacB:flΔ*rsf*S were mixed in 100 µL of LB medium at a ratio of 1:1, plated on LB agar medium, and then incubated for 10 h at 30 °C. To eliminate *E. coli* S17-1×pK18mobsacB:flΔ*rsf*S and *P. putida* PCL1760 which do not carry any vector, we resuspended the obtained bacterial cells in LB medium, plated on LB agar supplemented with ampicillin (100 µg/mL), and kanamycin (50 µg/mL), and then incubated at 30 ± 1 °C for 12 h. To get rid of the *P. putida* PCL1760×pK18mobsacB: flΔ*rsf*S, homogenous colonies were replated on LB plates supplemented with 10% sucrose and incubated for 12 h at 30 ± 1 °C. After incubation, two strains remained on agar plates (the *rsf*S gene deletion mutant and the original *P. putida* PCL1760). Further, single colonies were randomly selected, and a qualitative PCR *rsf*S gene was applied to obtain the mutant strain.

### 2.3. Mating Procedure and Generation of the SB9 (pJeM2:rsfS) Mutant

The mating procedure and generation of the SB9 (pJeM2:*rsf*S) mutant were carried out using the same procedure as in Section 2.2. To generate a DNA fragment containing structural gene *rsf*S with the following digestion sites, FauNDI and HindIII, a PCR was set with the primer rsfS-f/rsfS-r. The obtained fragment was digested with FauNDI and HindIII and ligated with the pJeM2 vector (digested with the same restriction enzymes). The resulting construct pJeM2:*rsf*S was then transferred into *E. coli* S17-1. Further, to obtain a complementation strain of SB9 (SB9’), the plasmid pJeM2:*rsf*S was conjugatively transferred from *E. coli* S17-1 into *P. putida* SB9.

### 2.4. Generation of Persister Cells of P. putida

#### 2.4.1. Preparation of Suspension Culture in Exponential Growth Cell Generation

The overnight LB medium cultures of PCL1760, SB9, and complementary recombinant of SB9 (SB9’) were inoculated in fresh LB medium to a final OD_595_ of 0.1 and incubated at 30 ± 1 °C with an agitation rate of 180 rpm/min. The exponentially-growing cell generation was determined by measuring the OD_595_ of the culture using a spectrophotometer (Spectrost-Nano BMG Labtech, Ortenberg, Germany).

#### 2.4.2. Induction of the *rsf*S Gene expression of PCL1760, SB9, and SB9’

The expression of the *rsf*S gene was induced by adding 0.2%(*w*/*v*) L-rhamnose to the cell cultures 30 min after the lag phase. The cultures were further reincubated to an optical density of 0.8 at 595 nm of their exponentially-growth phase. The resulting bacterial suspension was named PCL1760R (strain PCL1760 pretreated with L-rhamnose), SB9R (strain SB9 pretreated with L-rhamnose), and SB9’R (complementary recombinant of SB9 pretreated with L-rhamnose).

#### 2.4.3. The Persister Cells Generation

The comparison of persister cell generation between *P. putida* PCL1760, SB9, PCL1760’, and SB9’ was conducted according to the method described by Song and Wood [24] with modifications. To induce starvation, the antibiotic rifampicin was added to the culture suspensions (PCL1760, PCL1760R, SB9, SB9R, and SB9’R) at the exponential growth phase to a final concentration of 100 µg/mL as a transcription inhibitor and then incubated for 30 min. The culture suspensions were further centrifuged at 4000× *g* for 2 min at room temperature. The resulting pellet was resuspended in LB medium amended with ceftriaxone to a final concentration of 100 µg/mL (to lyse any non-persister cells) and incubated for 3 h. The suspensions were washed once with sterile phosphate-buffered saline (PBS) [140 mM NaCl, 5 mM KH_2_PO_4_, 1 mM NaHCO_3_, and pH 7.4], re-suspended in sterile, and resuspended in LB. The series of dilutions were then made in 96-well plates containing 200 µL of sterile LB medium by resuspending and transferring 15 times 50 µL of the suspension culture from well to well. Plates were incubated at 30 ± 1 °C for 2 days. The number of persister cells (PC) was calculated using the following formula:(1)$$P.C\;\left(\frac{\mathrm{CFU}}{\mathrm{mL}}\right)=a^{n}\times b$$
where a is the volume of suspension culture used, n is the number of overgrown wells, and b is the volume of nutrient broth added in each well.

### 2.5. Comparison of rsfS Gene Expression Profiles during Plant Root Colonization

#### 2.5.1. Bacterial Suspension Preparation

The bacterial suspension was prepared from the bacterial culture of *P. putida* PCL1760 grown in LB at 30 ± 1 °C for 12 h with a regular shaking speed of 150 rpm/min. The bacterial culture was centrifuged at 4000× *g* rpm for 5 min, at 4 °C with an agitation rate of 150 rpm/min. After that, the obtained pellets were washed twice with sterile PBS and resuspended in 1% PBS to a final optical density (OD) value of 0.1 at 595 nm.

#### 2.5.2. Plant Growth Condition

Tomato (*Solanum Lycopersicum*) seeds of the variety "Dubrava" (Shchyolkovo, Moscow, Russia) were sterilized according to Simons et al. [25]. After sterilization, seeds were placed on agar-solidified plant nutrient solution (PNS) [(g/L): 5.0 mM Ca(NO_3_)_2_ × 4H_2_O, 1.18 g; 5.0 mM KNO_3_, 0.5 g; 2.0 mM MgSO_4_ × 7H_2_O, 0.48 g; pantothenic acid solution; 1.0 mL and agar; 1.5%] and placed in the dark at 4 °C for 16 h, as a seed dormancy release with a subsequent incubation for 2 days at 28 ± 1 °C. The pre-sprouted tomato seeds were inoculated for 10 min in the cell suspension and sown in the gnotobiotic systems. The gnotobiotic systems were maintained in the climate control chamber under 90% light, with 16:8 day–night cycles, and 70% humidity. For statistical reliability, the experiment was triplicated and repeated twice. After 7 days, the basal and apical roots of the tomato plant were cut into 1 cm long pieces using a sterile scalpel. The pieces were then used to quantify the expression of the *rsfS* gene.

#### 2.5.3. Quantification of *rsf*S Gene Expression

The quantitative comparison of the RNA transcripts was performed using a one-step reverse-transcription quantitative polymerase chain reaction (qRT-PCR). For this purpose, the total RNA from root pieces was isolated using ExtractRNA reagent (Evrogen, Moscow, Russia) according to the manufacturer’s instructions. The qRT-PCR analysis was carried out in a 25 μL reaction system containing 5 μL of 5× one-tube qRT-PCR SYBR (Evrogen, Moscow, Russia), 0.4 μM of forward and reverse QrsfS primers, 1 μL of one-tube reverse transcriptase (Evrogen, Moscow, Russia), 0.5 μL of 50×SYBR Green I (Evrogen, Moscow, Russia), 2 μL of template nucleic acid (10 ng), and deionized water (nuclease-free). The experiment was performed in triplicates and repeated three times using a CFX96 Real-Time PCR System (Bio-rad, Hercules, CA, USA) under the following conditions: a 15 min reverse transcription step at 55 °C, a reverse transcriptase inactivation and polymerase activation at 95 °C for 1 min, followed by 40 cycles of 15 s denaturation at 95 °C, a primer annealing temperature at 60 °C for 20 s, and with an elongation temperature of 72 °C for 30 s. The final cycle was followed with a melting curve from 65 °C to 95 °C with a temperature increment of 0.5 °C each 0.5 s. The relative fold change of target gene expression was determined using the *rpoC* gene (gene encoding the β-subunit of DNA-dependent RNA polymerase) as an endogenous control based on the formula 2^−ΔΔCt^ [26].

### 2.6. Growth Rate Comparison of P. putida SB9 with PCL1760

For growth rate comparison, a bacterial suspension of *P. putida* SB9 and PCL1760 prepared as described in Section 2.4.1 was inoculated in a fresh LB broth medium and basal medium (g/L: 5.8 g K_2_HPO_4_; 3 g KH_2_PO_4_; 1 g NH_4_SO_4_ 0,2 g MgSO_4_ × 7H_2_O; and 0.5% of C_6_H_12_O_6_) to a final OD_595_ of 0.12. The cultures were incubated in three replicates using 96-well cell culture plates (Costar, New York, NY, USA) for 18 h at 30 ± 1 °C. The growth curve was estimated by measuring the OD of each culture once per well per hour at a longer wavelength (λ) of 595 nm using a spectrophotometer (Spectrostar Nano BMG Labtech, Ortenberg, Germany).

### 2.7. Comparison of the Colonization Ability of PCL1760 and Mutant SB9

#### 2.7.1. Colonization Ability of PCL1760 and Mutant SB9 as a Single Inoculum

The ability of SB9 to colonize tomato plants in comparison to its wild type strain PCL1760 was tested in gnotobiotic systems according to the method described in Section 2.5.2. For this purpose, tomato seeds were sterilized and inoculated with a bacterial suspension of wild type and SB9, which were prepared as described in Section 2.5.1 After cultivation, the basal and apical roots of the tomato plant were cut into 1 cm long pieces using a sterile scalpel and immersed in sterile PBS buffer for 15 min. A three-fold serial dilution was performed, after which 100 μL of 10^3^ dilutions were plated on LB medium. Plates were incubated overnight at 30 ± 1 °C and the colony-forming unit (CFU) per centimeter root length was calculated.

#### 2.7.2. Colonization Ability of PCL1760 and Mutant SB9 in Inoculum Combination

The colonization ability of SB9 and PCL1760 in consortium inoculum was also performed in gnotobiotic systems, according to the method described in Section 2.5.2. In this case, after sterilization, tomato seeds were inoculated with a bacterial mixture suspension of wild type and its mutant, which was prepared as described in Section 2.5.1. After cultivation, the basal and apical roots of the tomato plant were cut into 1 cm long pieces using a sterile scalpel and immersed in a sterile PBS buffer for 15 min. PBS was then used to quantify the total DNA concentration. The total DNA isolation was performed using Trizol reagent (Invitrogen, Waltham, MA, USA) according to the manufacturer’s instructions. Since the mutant strain does not carry the *rsf*S gene, a specific oligonucleotide forward primer (del-*rsf*S) was developed at the deleted gene site (Table 1, purpose: colonization ability).

Real-time PCR was performed using a CFX96 Real-Time PCR System (Bio-rad, Hercules, CA, USA) under the following conditions: denaturation and polymerase activation at 95 °C for 4 min followed by 40 cycles of 15 s denaturation at 95 °C, a primer annealing temperature of 60 °C for 20 s, and an elongation temperature of 72 °C for 30 s. The final cycle was followed with a melting curve from 65 °C to 95 °C with a temperature increment of 0.5 °C each 0.5 s. A pure DNA of the deletion *rsf*S site and *rsf*S gene fragment amplified from SB9 and PCL1760 were used to construct the calibration standard curve, respectively.

### 2.8. Biocontrol Properties of P. putida SB9 on the Tomato Plant against Forl ZUM2407

#### 2.8.1. Preparation of Conidial Suspension of Forl ZUM2407 

The conidial suspension of *Forl* ZUM2407 was prepared from a 5-day-old *Forl* ZUM2407 culture grown in potato dextrose broth medium (Merck, NJ, USA) at 25 ± 1 °C and an agitation rate of 150 rpm/min. To separate hyphae from spores, the culture was filtered using sterile cotton wool. The titer of the obtained suspension was counted using a hemocytometer and diluted with sterile PNS to a final density of 104 spores/mL.

#### 2.8.2. Biocontrol Ability of PCL1760 and Mutant SB9

The ability of *P. putida* SB9 to protect tomato plants against *Forl* ZUM2407, as compared with the wild type strain *P. putida* PCL1760, was carried out under laboratory conditions in pots containing mineral wool presoaked with a conidial suspension of *Forl* ZUM2407 to 60% of its water-holding capacity. The inoculum preparation of *P. putida* SB9 and *P. putida* PCL1760 was prepared as described above, with some modifications. In this case, after washing with sterile PBS, the obtained pellets were resuspended in 2% carboxymethylcellulose (CMC) to a final optical density (OD) value of 0.1 at 595 nm. Sterile tomato seeds were inoculated in each cell suspension for 25 min and dried in a laminar flow hood. Seeds were grown to 36.5 cm length × 13.5 cm width × 12 cm depth pots (70 seeds per pot in three repeats per treatment) (Appendix A). For the control group, seeds were inoculated in 2% CMC. Pots were then maintained for up to 28 days in the climate control chamber with the following setting conditions: temperature, 26 ± 1 °C; light cycle, 90% with 16:8 day–night cycles; and 70% humidity. The disease index (DI) was calculated according to the formula [27]:(2)$$\mathrm{DI}\left(\%\right)=\frac{\left[\left(n_{0}\times 0\right)+\left(n_{1}\times 1\right)+\left(n_{2}\times 2\right)+\left(n_{3}\times 3\right)+\left(n_{4}\times 4\right)\right]}{\begin{array}{c} \left(n_{0}+n_{1}+n_{2}+n_{3}+n_{4}\right) \end{array}}\times 100$$
where *n*_0_, *n*_1_, *n*_2_, *n*_3_, and *n*_4_ are the number of plants with, respectively, indices of 0, 1, 2, 3, and 4.

Statistical analysis of DI among treatments was carried out using originLab pro SR1b9.5.1.195 (OriginLab Corp., Northampton, MA, USA). The significant difference between groups was evaluated using one-way ANOVA and post hoc Tukey’s honestly significant difference test at *p*-value ≤ 0.05).

## 3. Results

### 3.1. Construction of Chromosomal rsfS Gene Deletion in P. putida PCL1760

The mutant SB9 was obtained after a horizontal transfer of the vector pK18mobsacB:flΔ*rsf*S from *E. coli* S17-1 to PCL1760 (Appendix A) after the subsequential growth on the selective LB agar medium with sucrose. Among the selected colonies after plating, seven colonies (E1–E7) out of thirteen selected colonies had a chromosomal deletion of the *rsf*S gene with fragments of approximately 300 bp. Experimental data of gel electrophoresis confirming the stages of genetic construction and the acquisition of the mutant is presented in Appendix A.

### 3.2. Comparison of rsfS Gene Expression Profiles during Plant Root Colonization

We compared the level of *rsf*S gene expression of *P. putida* PCL1760 on the apical and basal parts of the roots after their colonization of tomato plants. The obtained results are shown in Figure 3. A statistical difference at a *p*-value ≤ 0.01 of the relative gene expression was observed. The level of relative *rsf*S gene expression on the basal plant roots was 2.48 times less than on the apical plant roots. Their relative Ct values were 1.02 ± 0.18 and 0.41 ± 0.13, respectively.

### 3.3. Growth Rate Comparison of P. putida SB9 with PCL1760

The growth curve of SB9 and PCL1760 is represented in Figure 4A (in a rich nutrient medium, LB) and Figure 4B (basal medium). The obtained result revealed that the growth rate of SB9 was almost identical to PCL1760 in both rich and basal media. The lagging growth phase of SB9 and PCL1760 was immediately observed after two hours of incubation in a rich LB medium (Figure 4A) and four hours in a basal medium (Figure 4B). Their average optical density in the lagging phase was 0.15 ± 0.02 and 0.14 ± 0.02 in the LB medium, and 0.15 ± 0.025 and 0.14 ± 0.015 in the basal medium. However, after six hours of incubation (the beginning of the stationary phase), a statistical difference (*p* ≤ 0.05) between SB9 and PCL1760 was observed (Figure 4A). The OD_595_ of SB9 during this incubation period ranged from 0.976 ± 0.013 to 1.22 ± 0.02, while the OD_595_ of PCL1760 reached from 0.85 ± 0.02 to 1.13 ± 0.03.

### 3.4. A Comparison of the Colonization Ability of PCL1760 and Mutant SB9

The ability of SB9 compared to its wild type strain PCL1760 to colonize the roots of tomato plants is shown in Figure 5 (inoculated separately) and Figure 6 (combined). The results revealed that in a non-competitive environment, the colonizing ability of the mutant statistically (*p* ≤ 0.05) did not differ from its wild type strain PCL1760 (Figure 5). The average population was almost at the same level on the basal and apical parts of the root. The colony-forming unit of SB9 and PCL1760 per centimeter (CFU/cm) root length on the basal part was 2.82 × 10^3^ ± 0.15 × 10^3^ and 2.88 × 10^3^ ± 0.08 × 10^3^ whereas, on the apical part of the root, the population was counted as 8.78 × 10^3^ ± 0.6 × 10^3^ and 8.73 × 10^3^ ± 0.9 × 10^3^, respectively.

On the other hand, in competitive colonization, a mutant devoid of the *rsf*S gene was significantly inferior to its wild type strain (Figure 6). The obtained PCR result revealed a 4-fold increment of PCL1760 DNA concentration on both the basal and apical parts of the tomato root in comparison to SB9. The DNA concentration of SB9 on the basal and apical part of the plant was assayed as 3.94 × 10^3^ ± 9.3 × 10^2^ and 3.01 × 10^3^ ± 7.02 × 10^2^ fg/µL, respectively, whereas the DNA concentration of PCL1760 was evaluated as 9.04 × 10^3^ ± 2.8 × 10^3^ fg/µL and 2.14 × 10^4^ ± 6.7 × 10^3^ fg/µL on the basal and apical part, respectively.

### 3.5. The Generation of Persister Cells of P. putida 

The obtained result revealed that the *rsf*S gene deletion mutant *P. putida* SB9 is less persistent in comparison to its wild strain PCL1760 (Figure 7). Moreover, under starvation conditions, the complementary recombinant L-rhamnose-treated bacterial cells of SB9 (SB9’R) were found to be more persistent to the antibiotic ceftriaxone.

The remaining cell subpopulation of SB9 was up to 221.07 times less than SB9’R. The generation of the persistent cell of PCL1760, the L-rhamnose pretreated strain PCL1760 (WTR), the L-rhamnose pretreated strain SB9 (SB9R), and the complementary strain of SB9 (SB9’) was individually up to 64.43, 43.15, 143.19, and 43.15 times less persistent than SB9’R, (Figure 7). The colony-forming units per milliliter of PCL1760, PCL176R, SB9, SB9R, SB9’, and SB9’R after starvation were estimated as 1.82 × 10^6^, 2.72 × 10^6^, 5.31 × 10^5^, 8.2 × 10^5^, 2.72 × 10^6^, and 1.17 × 10^8^, respectively.

### 3.6. Biocontrol Properties of P. putida SB9

The biocontrol ability of *P. putida* SB9 showed significant suppression of TFRR disease caused by *Forl* ZUM2407 (Figure 8), as compared with plants without treatment (*Forl* group). The disease progression in the group of plants pretreated with the *rsf*S gene deleted mutant SB9 was 19.03 ± 1.58%, whereas the disease progression of plants without treatment was 27.28 ± 2.68%.

Moreover, a statistically significant difference (*p* ≤ 0.05) in the disease progression between groups pretreated with *P. putida* SB9 and the original strain of *P. putida* PCL1760 was observed (Figure 8). The DI of plants pretreated with the original strain PCL1760 was recorded as 12.20 ± 1.55%. The effectiveness of *P. putida* SB9 and PCL1760 to inhibit the growth of *Forl* ZUM2407 was calculated as 30.24 ± 5.79% and 55.28 ± 5.68%.

## 4. Discussion

*Pseudomonas* strains as biocontrol agents are known to be better colonizers in comparison to their *Bacillus* counterparts, although several strains of both genera are known to produce secondary metabolites that are involved in direct antibiosis of phytopathogenic fungi taking into consideration *P. putida* [2,28,29]. Among biocontrol agents, *Pseudomonas* strains are known to be of best colonizers due to their developed motility apparatus and wide range of utilized substrates, covering all major compounds of plant root exudates [30]. Both abilities–motility and efficient nutrient consumption are constituents of the CNN mechanism of biocontrol, referred also as competitive colonization [4]. Under natural conditions, root plants are colonized with a variety of microorganisms, which differ greatly in motility, and a set of consumed compounds [31]. This might imply persistence as an additional important constituent of the competitive colonization mechanism.

Persistence to environmental stresses in eubacteria is governed by a stringency response system. Being part of the stringency response ribosome hibernation genes are frequently located within starvation operons and are mostly expressed in the stationary phase of bacterial growth [32,33]. The ribosome silencing factor (*rsfS*) is a conserved gene found in bacteria and eukaryotic organelles [14]. Based on the data published on the mechanism of hibernation, RsfS is a key gene in ribosomal hibernation [16,34,35], and the reason that it was chosen was to confirm persistence in our root colonizer *P. putida* PCL1760. Recent work on PCL1760 showed its susceptibility to ceftriaxone, which inhibits cell wall synthesis [9,36]. Thus, in this study, only dividing cells are susceptible to death in the presence of the antibiotic. We, therefore, suggest persistence as an important phenotypic trait of a good colonizer to protect plants from xenobiotics. We associate this with “survival of the fittest” by the ability to conserve energy through hibernation under stressful conditions since root exudates are being released periodically from the tip of the root and less at the base. The growth dynamic of SB9 and PCL1760 practically did not differ in both minimal and nutrient media except for the stationary phase in the LB medium. We suggest that the deletion of the *rsf*S gene in SB9 did not impair its growth in both media (Figure 4). A similar result was observed when the two strains were separately inoculated on tomato seedlings where the CFU of both strains statistically did not differ (Figure 5).

To evaluate the importance of the *rsf*S gene, we determined its relative expression in bacterial cells on the roots of tomato plants. The results showed expression being higher at the root tip than at the basal part, although vice versa was expected. We suggest that translation of *rsf*S mRNA had already occurred at the basal part with most cells in hibernation. Therefore, the detection was low in comparison to the apical part where bacterium cells of PCL1760 might be at the end of the exponential stage (Figure 4). The synthesis of mRNA is said to reduce under nutrient starvation conditions and, therefore, the expression and translation of these genes would have taken place earlier before starvation at the basal point of the root [37]. Additionally, this is due to the dynamics in the stability of mRNA, whereby its degradation rate is indirectly proportional to the growth rate of the bacterium *Lactococcus lactis* due to progressive isoleucine starvation [38]. Since RsfS is involved in one of the initiation stages of ribosome disassociation after translation, we suggest a complete translation of the protein itself has occurred and its mRNA decayed during starvation; therefore, the lower expression was at the basal point [34].

A state of dormancy triggered by hibernation factors results in the persistence of bacteria cells to withstand stressful conditions without mutations [39]. As proposed by the functions of hibernation factors in the reduction in bacterial sensitivity to different antibiotics, the persistence of the wild type strain PCL1760 was confirmed, as reported by McKay and Portnoy [12]. The authors reported on the susceptibility of *E. coli* to the antibiotic gentamicin after deletion of the ribosome modulation factor (rmf), another hibernation factor, and also a high tolerance by a hibernation-promoting factor (hpf) gene to aminoglycosides, except to beta-lactam or quinolone antibiotics [12]. In our study, PCL1760 was much more persistent in comparison to SB9, where rifampicin, being an inhibiter to RNA polymerase, initiated the expression of rsfS that increased the tolerance to the subsequent treatment of the cephalosporin antibiotic (β-lactam) and ceftriaxone acting during cell deletion. Complementary SB9 cells with the plasmid after activation of the *rsf*S gene showed extremely higher persistence than the wild type. This is due to the induced episomal expression of the *rsf*S by rhamnose promoter *E. coli rha*P*_BAD_* in pJem2 [21]. We suggest that RsfS is responsible for the high tolerance of PCL1760 to the β-lactam antibiotic ceftriaxone, as RMF and HPF are to aminoglycosides [12,24].

The colonization ability of PCL1760 and its mutant (SB9) on the root tomato plants showed the dominance of PCL1760 on the root of tomato seedlings with a significant difference in both basal and apical parts. As hypothesized, this ability correlated with the biocontrol ability of the two strains acting through the mechanism of CNN against *Forl* ZUM2407. Although SB9 was able to control TFRR disease, PCL1760 with much persistence was able to control double the margin. In all experimental setups, deletion of the *rsf*S gene did not affect other phenotypic properties of the strain PCL1760 but reduced colonization ability and effective control of diseases. Thus, persistence based on one of the important genes for hibernation (*rsf*S) showed this trait as an important factor for bacterial biocontrol agents acting through the mechanism of CNN.

## 5. Conclusions

The phenotypic characteristics of good bacterial colonizers involved in the CNN mechanism of biological control agents are normally based on the mobility of a strain and the presence of root exudates. We proposed an additional characteristic, the persistence of a bacterial strain as a key factor to consider when selecting good root colonizers acting solely through CNN. Since hibernation is an important factor for persistence, the gene *rsf*S being an important hibernation factor in bacteria was able to confirm this characteristic in the *P. putida* PCL1760 model. By knocking out the gene in the PCL1760 mutant, SB9 had a reduction in persistence, colonization, and biocontrol abilities in direct contact with its wild type strain. This competition occurs when pathogenic strains with more persistence turn to dominate good root colonizers, thereby reducing their effectiveness. We deduce from this study that quantitative expression analysis of the *rsf*S gene can be an indicator for the selection of good colonizers for microbes protecting plants via CNN mechanism. Further studies into different hibernation genes concerning persistence may broaden the knowledge into other mechanisms of biological control agents.

## Figures and Tables

**Figure 1 microorganisms-11-00019-f001:**
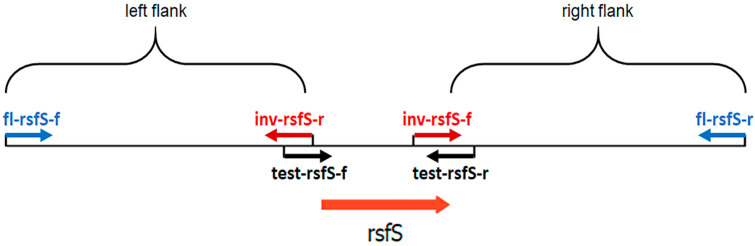
The layout of primers used in this study. fl-rsfS-f/fl-rsfS-f (blue). fl-rsfS fragments primers and inv-rsfS-f/inv-rsfS-r (red), the primers for pUC19 synthesis. flΔrsfS, and test-rsfS-f/test-rsfS-r (black), the primers for test PCR. Arrows indicate the direction of nucleotide chain synthesis.

**Figure 2 microorganisms-11-00019-f002:**
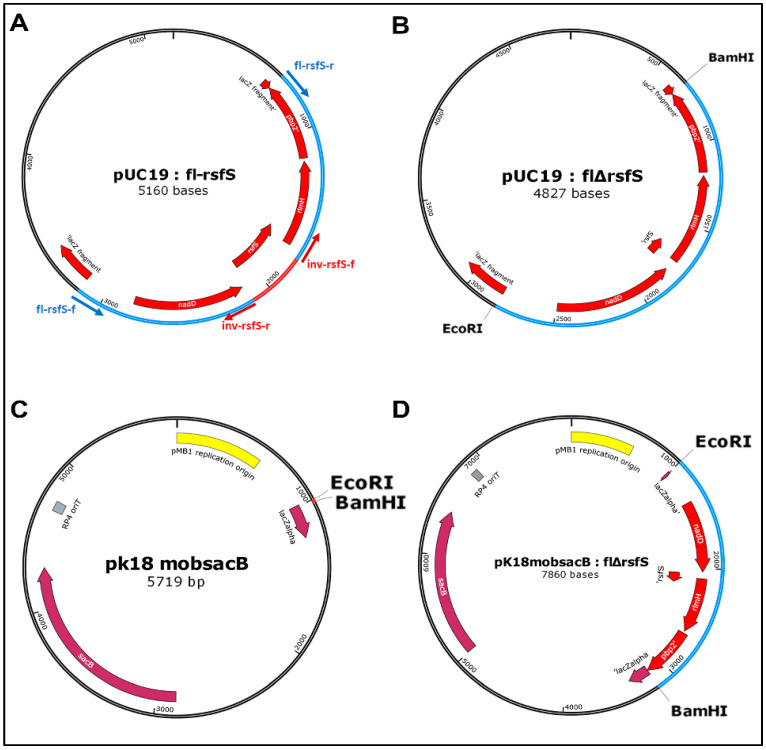
Molecular cloning steps of *rsf*S gene deletion, where (**A**) is the pUC19 vector with the fl-rsfS insert (blue), (**B**) is the pUC19 vector after deletion of the *rsf*S gene with the preservation of flanking sequences, (**C**) is the pK18mobsacB vector with EcoRI and BamHI restriction sites, and (**D**) is the pK18mobsacB vector with flΔ*rsf*S insertion.

**Figure 3 microorganisms-11-00019-f003:**
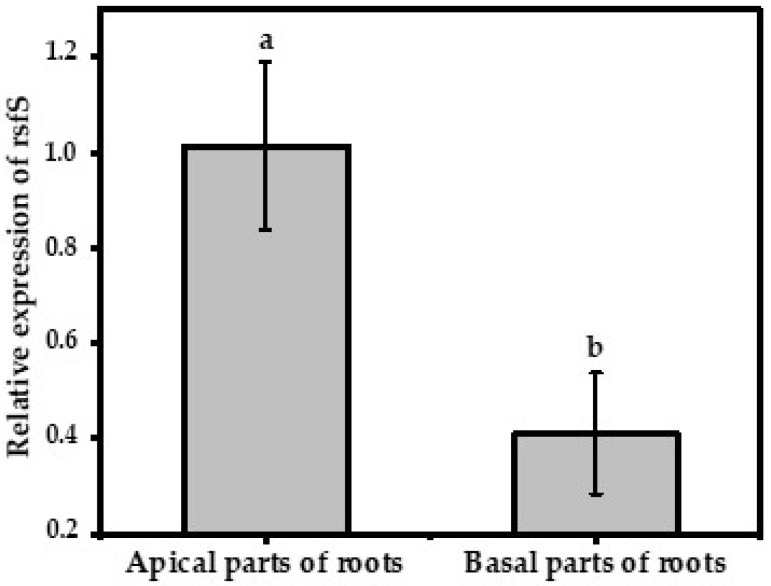
An RT-qPCR-based comparison of the relative *rsf*S gene expression in *P. putida* PCL1760 under the colonization of tomato root parts. Data are represented as mean ± SD. The significant difference between bacterial strains was evaluated using one-way ANOVA and post hoc Tukey’s honestly significant difference test at a *p*-value ≤ 0.01. Significant difference among groups is labeled with different letters (a and b).

**Figure 4 microorganisms-11-00019-f004:**
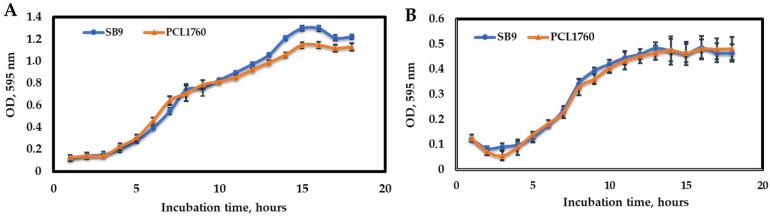
Growth curve comparison of *P. putida* SB9 and PCL1760 in a rich LB medium (**A**) and a basal medium (**B**). Data are represented as mean ± SD.

**Figure 5 microorganisms-11-00019-f005:**
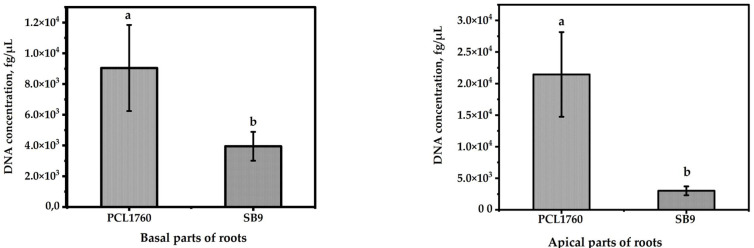
Colonization ability of *P. putida* SB9 compared with PCL1760. Data are represented as mean ± SD. The significant difference between bacterial strains was evaluated using one-way ANOVA and post hoc Tukey’s honestly significant difference test at a *p*-value ≤ 0.01. Significant difference among groups is labeled with different letters (a and b).

**Figure 6 microorganisms-11-00019-f006:**
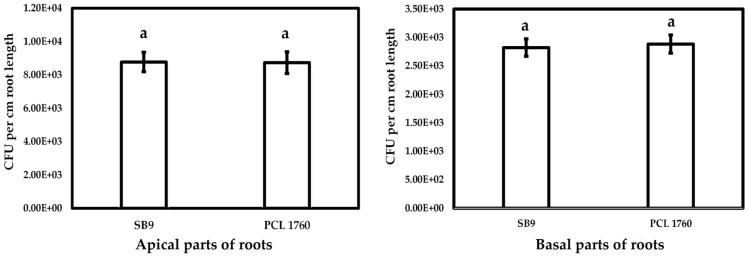
Colonization ability of *P. putida* SB9 compared with the strain of PCL1760. Data are represented as mean ± SD. The significant difference between bacterial strains was evaluated using one-way ANOVA and post hoc Tukey’s honestly significant difference test at *p*-value ≤ 0.01. Significant difference among groups is labeled with letter (a).

**Figure 7 microorganisms-11-00019-f007:**
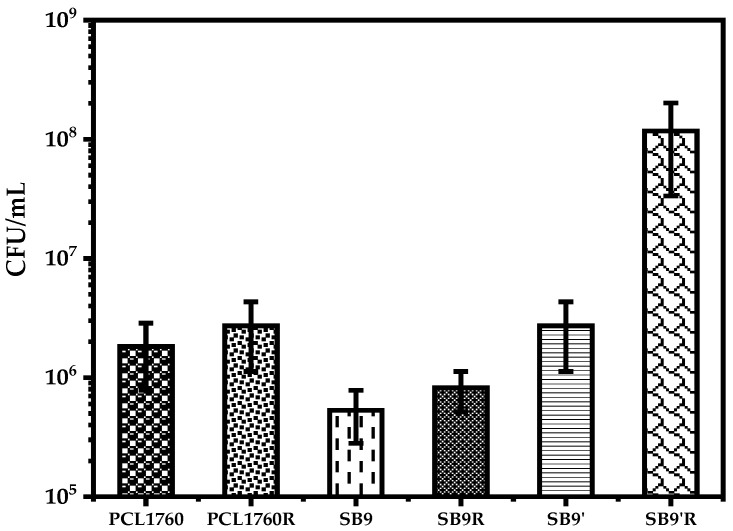
Generation of persister cells of PCL170 strains. PCL1760, PCL1760R, SB9, SB9R, SB9’, SB9’R as a wild type PCL1760, a wild type PCL1760 pretreated with L-rhamnose, the *rsf*S gene deleted mutant of PCL1760, SB9 pretreated with L-rhamnose, the complementary recombinant of SB9, and complementary recombinant of SB9 pretreated with L-rhamnose, respectively. Data are represented as mean ± SD.

**Figure 8 microorganisms-11-00019-f008:**
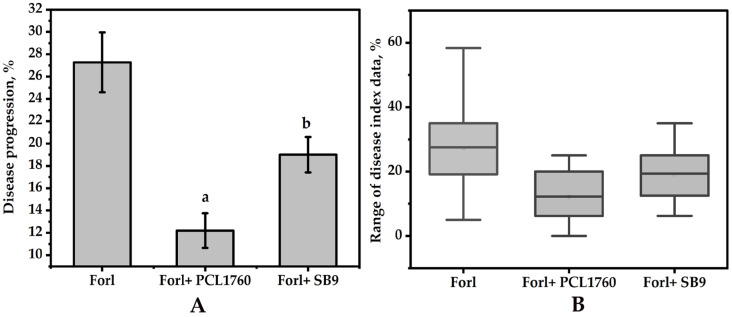
Biocontrol properties of *P. putida* SB9 in the suppression of tomato foot and root disease caused by *Forl* ZUM2407. (**A**) Disease index of each group (data are represented as mean ± SD) and (**B**) the range of the disease index data of tomato plants. Significant difference among groups at a *p*-value ≤ 0.05 is labeled with different letters (a and b).

**Table 1 microorganisms-11-00019-t001:** The microbial strains used in this study.

Strain, Plasmid	Genotype/Phenotype/Description	Source, Reference
**Strains**		
*E. coli* DH5α		Invitrogen, Cat. No. 12297–016
*E. coli* S17-1		Simon et al. [18]
*P. putida* PCL1760	Efficient root colonizer isolated from avocado rhizosphere	Validov et al. [7]
*P. putida* SB9	*P. putida* PCL1760 (Δ*rsf*S) obtained	This work
*P. putida* SB9 (pJeM2:*rsf*S)	Mutant SB9, harbouring pJeM2: *rsf*S plasmid	This work
*Fusarium oxysporum* f.sp. *radices-lycopersici* ZUM2407	Tomato root rot pathogen	Bolwerk et al. [19]
pK18SacBmob	KmR, integration vector for pseudomonads	Schafer et al. [20]
**Plasmids**		
pK18SacBmob:flanks-Δ*rsf*S		This work
pJeM2	Broad host range vector, Km	Jeske and Altenbuchner [21]
pJeM2:*rsf*S		This work

**Table 2 microorganisms-11-00019-t002:** Oligonucleotide primers used in this study.

Purpose	Description	Primers	Sequence	Reference
The *rsf*S gene deletion of *P. putida* PCL1760	The *rsf*S gene fragment with flanking sequences	fl-rsfS-f	CCGAAGCCGACTGGCACACG	In this study
		fl-rsfS-r	GTGTCGACCACGCCGTAATGGG	
	Inverse *rsf*S gene primer (inv- *rsf*S)	inv-rsfS-f	CTACGACCTTGAGCGTCTGTGGC	In this study
		inv-rsfS-r	GACTGCCACGGAGACCGCG	
	*rsf*S gene	test-rsfS-f	AGCGGCAAATCGGTGAGGTTCCTG	In this study
		test-rsfS-r	GTACTTATACGGTTCGCGGACGGG	
Quantification of the *rsf*S gene expression	Housekeeping genes	rpoC-f	CAAGCGTCTGAAGCTGATGGAAGC	In this study
		rpoC-r	GGAAGTCGCGAAACGGCCACC	
	*rsf*S gene	QrsfS-f	GATGACCGCCGCTGCCCG	In this study
		QrsfS-r	AGCGCTTTCCTTATTCGCGGTCTTTG	
Colonization Ability	Mutant of PCL1760	delrsfS-f	GGTGCCTCTGGCGCCTACG	In this study
		QrsfS-r	AGCGCTTTCCTTATTCGCGGTCTTTG	
	Wild type strain of PCL1760	QrsfS-f	GATGACCGCCGCTGCCCG	In this study
		QrsfS-r	AGCGCTTTCCTTATTCGCGGTCTTTG	
The *rsf*S gene cloning in pJeM2	*rsf*S gene	rsfS-f	TTTTTT**CATATG**ACCAAGCAGAAAATTTACGGCG	In this study
		rsfS-r	TTTTTT**AAGCTT**ATTCGCGGTCTTTGAGCTTGTC	

## Data Availability

Not applicable.

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
