# Peer review of "Persistence as a Constituent of a Biocontrol Mechanism (Competition for Nutrients and Niches) in Pseudomonas putida PCL1760"

_microorganisms, 2022, doi:10.3390/microorganisms11010019_

Round 1

Reviewer 1 Report

The research paper investigates the role of persistence in biocontrol strategy involving plant-microbe interactions and whether it could be considered as a relevant indicator (additionally to CNN) for the biocontrol efficiency. The study presented by the author is novel and highly interesting. The experimental design is well done, and the data are well presented.

Why the In-vitro confrontation assay (bacteria VS fungal pathogen) has not been done? This could be interesting to compare the direct effect of the bacterial strains on some parameters such as conidial germination and mycelium growth    

More data needs to be provided with respect to the agronomic trials, namely the fertilization regime and pot size. 70 seeds per pot seems exaggerated, what justify this huge number?  

Analyzing some plant antioxidant parameters could be a good addition to see how the infection translates at the physiological level.

Experiment was carried out in a soilless substrate. I understand that such choice is justified by the specific objective of the study. Though evaluating the efficiency of the mutant and non-mutant strain in non-sterile soil (where there would be microbial competition) might be required to validate the hypothesis. This could be adressed in the discussion section.    

Author Response

Dear Reviewer,

Thank you for taking the time to review our work. We really appreciate all of your hard work done in generating a detailed peer-review of our manuscript. Kindly find below our response to your comments.  All changes in the initial version of the manuscript are in red font color for added sentences and strike-through, and highlighted in yellow for deleted words. We have answered all the comments, and the responses are attached to this letter.

Reviewer 2 Report

Thank you very much for the opportunity to review the paper entitled Persistence as a constituent of biocontrol mechanism (competition for nutrients and niches) in Pseudomonas putida PCL1760. I have to underline that I have not read such well-written paper for a long time. The topic is interesting and the importance of the presented results is very high. All conducted experiments are explained in details and the results are presented in the scientific manner.

My only concern is related to the used literature. Namely there is numerous references older than a decade. I suggest to fortify this extraordinary work with the newest research in the field.

Also, please, change commas with dots as a decimal place in the figures throughout the manuscript.

Author Response

Dear Reviewer,
Thank you for taking the time to review our work. We really appreciate all of your hard work done in generating a detailed peer-review of our manuscript. Kindly find below our response to your comments. All changes in the initial version of the manuscript are in red font color for added sentences and strike-through, and highlighted in yellow for deleted words. We have answered all the comments, and the responses are attached to this letter.

We hope the revised manuscript will be accepted for publication.
